# The Association between High-Sensitivity C-Reactive Protein and Metabolic Syndrome in an Elderly Population Aged 50 and Older in a Community Receiving Primary Health Care in Taiwan

**DOI:** 10.3390/ijerph192013111

**Published:** 2022-10-12

**Authors:** Yu-Lin Shih, Yueh Lin, Jau-Yuan Chen

**Affiliations:** 1Department of Family Medicine, Chang-Gung Memorial Hospital, Linkou Branch, Taoyuan 33305, Taiwan; 2College of Medicine, Chang Gung University, Taoyuan 33302, Taiwan

**Keywords:** high-sensitivity C-reactive protein, metabolic syndrome, middle-aged, elderly

## Abstract

Metabolic syndrome (MetS) has become the most important issue in family medicine and primary care because it is a cluster of metabolic abnormalities that are a burden on health care in many countries. Highly sensitive C-reactive protein (hsCRP), which is elevated in inflammatory situations, can be produced by monocyte-derived macrophages in adipose tissue. People with MetS tend to have more adipose tissue. Therefore, we aimed to investigate the association between hsCRP and MetS among elderly individuals aged 50 years and older in northern Taiwan. This study was a cross-sectional community-based study that included 400 middle-aged and elderly Taiwanese adults, and 400 participants were eligible for analysis. We divided the participants into a MetS group and a non-MetS group. Pearson’s correlations were calculated between hsCRP and other related risk factors. Furthermore, the relationship between hsCRP and MetS was analyzed with logistic regression. People in the MetS group were more likely to have higher hsCRP levels. The Pearson’s correlation analysis showed a positive correlation with hsCRP. In the logistic regression, hsCRP was significantly associated with MetS, even with the adjustment for BMI, uric acid, age, sex, smoking status, drinking status, hypertension, diabetes mellitus, and dyslipidemia. In summary, our research indicated that hsCRP could be an independent risk factor for MetS.

## 1. Introduction

With wealthy lifestyles and longevity, metabolic syndrome (MetS) has become the most important issue in family medicine and primary care because it is a cluster of metabolic abnormalities, such as overweight, glucose intolerance, elevated blood pressure (BP) and dyslipidemia [1], leading to a higher risk of diabetes mellitus (DM) and cardiovascular disease [2,3]. Moreover, people with MetS tend to have more adipose tissue, resulting in the excessive release of free fatty acids, thereby reducing the peripheral insulin sensitivity and causing insulin resistance (IR) [4]. Expanded adipose tissue also leads to the overproduction of proinflammatory cytokines, including C-reactive protein (CRP), interleukin-6 (IL-6) and tumor necrosis factor alpha (TNF-α), by monocyte-derived macrophages in adipose tissue [5,6]. In addition to IR, inflammation appears to be a pathophysiologic phenomenon of MetS. Several studies have shown that high-sensitivity CRP (hsCRP) is positively associated with fasting insulin, IR and MetS [7,8,9,10]. Similar correlations have been found in different age groups (children and adults) [11,12,13]. Previous studies have also indicated that hsCRP positively correlates with many chronic diseases, such as hypertension (HTN), DM and dyslipidemia [14,15,16]. Lifestyle can also have an impact on hsCRP levels [17]. Conversely, the prevalence of chronic diseases and metabolic syndrome increases as people age [18]. Furthermore, lifestyle factors, such as smoking and drinking, can elevate the hsCRP level [17]. Elevated hsCRP has seemed to be impacted by multiple factors. However, there has been a lack of studies that have considered comprehensive parameters in the relationships between hsCRP and metabolic syndrome. In our research, we gathered many factors, from laboratory data to anthropometric parameters, to evaluate the association between hsCRP levels and metabolic syndrome among elderly individuals aged 50 years and older in northern Taiwan. The results of our research could provide a reference for primary care providers surveying metabolic syndrome.

## 2. Materials and Methods

### 2.1. Study Design and Study Population

This study was a cross-sectional community-based study designed to examine possible independent predictors of MetS. Participants were recruited from a community health promotion project of Linkou Chang Gung Memorial Hospital between 1 February 2014 and 31 August 2014. The project enrolled participants aged 50 years or older through posters or notifications from the community office. Each participant completed a questionnaire that included personal information and a medical history in a face-to-face interview. Anthropometric measurements were obtained, and blood sampling was performed by trained research assistants or nurses under the supervision of a medical doctor. The project was approved by the Institutional Review Board of Linkou Chang Gung Memorial Hospital, and all participants provided written informed consent prior to enrollment. We excluded participants based on the following exclusion criteria: (1) disability; (2) declining of participation; and (3) acute illness at enrollment or recently. Finally, a total of 400 participants were included in the analysis.

### 2.2. Measurements

Trained assistants or nurses collected personal information, including age, gender, smoking habit (self-reported current smoker or not), drinking status (drinking ≥ 2 days/week or not) and medical history, during a face-to-face interview. Anthropometric data, such as height, weight, waist circumference (WC) and BP, were measured. WC was measured at the level midway between the iliac crest and the lower border of the 12th rib while the subject stood with their feet 25–30 cm apart. BP was measured using an automated sphygmomanometer placed on the participant’s right arm after a 10-minute rest in a seated position, and the lowest reading was recorded. Body mass index (BMI) was calculated as the person’s weight in kilograms divided by the square of the height in meters. Participants were requested to fast for a minimum of 12 h and to avoid a high-fat diet or alcohol consumption for at least 24 h prior to blood sampling. Venous blood samples were obtained between 7 am and 11 am and stored in a refrigerator at 4 °C before analysis in the hospital laboratory. Clinical biochemical tests included hsCRP, fasting plasma glucose (FPG), total cholesterol (Total-C), triglyceride (TG), high-density lipoprotein cholesterol (HDL-C), low-density lipoprotein cholesterol (LDL-C) and uric acid (UA). Blood samples were collected in a hospital laboratory accredited by the College of American Pathologists (CAP). Trained clinical laboratory technicians performed all assessments. All information was entered into a centralized electronic database that was under strict quality control and monitored on a regular basis.

### 2.3. Definition of Metabolic Syndrome and Other Diseases

MetS was defined according to the modified criteria of the National Cholesterol Education Program Adult Treatment Panel III (NCEP-ATP III) [19], but the standard of WC has been adjusted by the Minister of Health and Welfare of Taiwan. MetS is defined by the presence of three or more of the following components: (1) WC ≥ 90 cm for men and ≥80 cm for women; (2) TG ≥ 150 mg/dL; (3) HDL-C < 40 mg/dL for men and <50 mg/dL for women; (4) BP ≥ 130/85 mm Hg or current use of antihypertensive medications; and (5) FPG ≥ 100 mg/dL. The participants who met the criteria of MetS were classified in the metabolic group, whereas the participants who did not meet the criteria were classified in the non-metabolic group. Hypertension (HTN) was defined as systolic BP (SBP) ≥ 140 mm Hg, diastolic BP (DBP) ≥ 90 mm Hg, or the use of medications for HTN. Diabetes mellitus (DM) was defined as fasting plasma glucose ≥ 126 mg/dL or the use of oral hypoglycemic agents or insulin. Dyslipidemia was defined as LDL-C ≥ 130 mg/dL and HDL-C < 40 mg/dL in men, HDL-C < 50 mg/dL in women, TG ≥ 150 mg/dL, total cholesterol ≥ 200 mg/dL, or the use of lipid-lowering medications. According to the Ministry of Health and Welfare of Taiwan, obesity was defined as BMI ≥ 27 kg/m^2^ [20]. Based on the Centers for Disease Control and the American Heart Association, elevated hsCRP was defined as plasma hsCRP ≥ 1 mg/L for a higher risk of cardiovascular disease [21,22].

### 2.4. Statistical Analysis

The participants were divided into two groups according to the metabolic syndrome. In Table 1, the categorical variables are expressed as n (%) and were analyzed using the chi-square test. The normality of the continuous variables was checked by the Shapiro-Wilk normality test. Continuous variables are presented as the mean ± [SD] if the variables (age, WC, BMI, SBP, DBP, HDL-C, LDL-C, total C and uric acid) were consistent with normally distributed variables and as the median [Q1, Q3] if the variables (hsCRP, FPG and TG) deviated significantly from a normal distribution. We obtained p values from the independent sample t test for data consistent with a normal distribution and the Mann-Whitney U test for data consistent with a nonnormal distribution. Pearson’s correlation coefficient was calculated to analyze correlations between age, WC, BMI, SBP, DBP, FPG, HDL-C, TG, LDL-C, Total-C and uric acid, as shown in Table 2. In the multivariate analysis shown in Table 3, a binary logistic regression was used to evaluate the relationship between metabolic syndrome and hsCRP level with adjustment for BMI, uric acid level, age, sex, smoking status, drinking status, HTN, DM and dyslipidemia. All tests were two-sided, and a p value of less than 0.05 was considered statistically significant. Data were analyzed by SPSS Statistics software, version 22 (IBM, SPSS Armonk, IMM Co, Armonk, NY, USA).

## 3. Results

This study recruited 619 participants through posters and notifications from the community office. A total of 219 subjects with incomplete data, with disabilities, refusing participation, or with acute illnesses at enrollment or recently were excluded. The final number of participants was 400. Table 1 shows the general characteristics of the participants. There were 141 men (35.30%) and 259 women (64.70%), with a mean age of 64.47 years. The mean SBP was 129.50 mm Hg, and the mean DBP was 76.93 mm Hg. The mean WC and BMI were 85.07 cm and 24.55 kg/m^2^, respectively. A total of 10.80% of the participants smoked, and 19.50% drank alcohol frequently. Overall, 50.39%, 19.80% and 65.00% of the participants had HTN, DM and dyslipidemia, respectively. The results of clinical biochemical tests showed that the average levels of hsCRP, FPG, total C, TG, HDL-C, LDL-C and UA were 1.28, 91.00, 197.15, 107.00, 54.43, 118.37 and 5.75 mg/dL, respectively. According to the definition of NCEP-ATP III, we divided the participants into a non-MetS group and a MetS group. The average levels of WC, BMI, SBP, DBP, hsCRP, FPG, TG and UA in the MetS group were significantly higher than those in the non-MetS group. Additionally, the MetS group had a significantly larger proportion of HTN, DM and dyslipidemia. There were no statistically significant differences between the groups in terms of age, sex, smoking, LDL-C or total C. Participants in the non-MetS group tended to have a higher HDL-C.

Table 2 demonstrates the correlations between baseline characteristics and hsCRP using Pearson’s correlation analysis. The results show that hsCRP was only negatively associated with HDL. The correlations of hsCRP with age, WC, BMI, SBP, DBP, FPG, TG, LDL-C, total C and uric acid were not significant.

In Figure 1, the elevated hsCRP was defined as hsCRP ≥ 1 mg/L, and we observed a higher prevalence of MetS in the elevated hsCRP group. The prevalence of MetS in the normal hsCRP group and elevated hsCRP group were 21.7 and 43.2, respectively, with a *p*-value < 0.001. Table 3 shows the results of the binary logistic regression analysis of the relationships between cardiometabolic risk factors and MetS. Traditional cardiometabolic risk factors, such as obesity, HTN, DM, dyslipidemia, age, sex, smoking, drinking, hsCRP and uric acid, were included in the multivariate analysis. Obesity was defined as BMI ≥ 27 kg/m^2^, and elevated hsCRP was defined as hsCRP ≥ 1 mg/L. In the univariate logistic regression, elevated hsCRP, obesity, uric acid, drinking status, HTN, DM and dyslipidemia were significantly associated with MetS. Furthermore, the multivariate logistic regression showed that elevated hsCRP (*p* = 0.01, OR: 2.24, 95% CI: 1.23–4.08), uric acid (*p* = 0.05, OR: 1.22, 95% CI: 1.00–1.48), drinking status (*p* = 0.04, OR: 0.46, 95% CI: 0.22–0.98), HTN (*p* < 0.001, OR: 4.92, 95% CI: 2.90–8.34), DM (*p* < 0.001, OR: 5.88, 95% CI: 3.13–11.05) and dyslipidemia (*p* < 0.001, OR: 3.26, 95% CI: 1.83–5.81) remained significantly associated with MetS.

## 4. Discussion

The criteria for MetS were WC, BP, FPG, HDL-C and TG [23]. Significant differences in these criteria between the nonmetabolic group and metabolic group were observed in our study. Previous studies have reported that serum uric acid was positively related to MetS [24,25], and our results in Table 1 also corresponded with previous studies. There was a high prevalence of DM, HTN and dyslipidemia in the MetS group, as also indicated by former studies [2,3]. Furthermore, hsCRP showed a significant, positive relationship with MetS, and this finding led us to speculate about the association between hsCRP and MetS.

In MetS, WC, SBP, DBP, FPG and TG are risk factors. HDL-C acts as a protective factor against MetS. Several studies have shown significant associations of hsCRP with the components of MetS [26,27,28]. We also observed a similar trend in the correlation between hsCRP and MetS. In Table 2, Pearson’s correlation analysis indicated significant correlations between hsCRP and MetS criteria, while a negative correlation was found between hsCRP and HDL-C. Because hsCRP shared the same risk and protective factors with MetS in our study, this finding raised the question of whether hsCRP could be an independent risk factor for MetS.

Figure 1 showed that the prevalence of metabolic syndrome in the elevated hsCRP level group was significantly higher than in the low hsCRP level group. Table 3 shows the logistic regression analysis. In Model 1, the univariate logistic regression revealed positive relationships of MetS with hsCRP levels, obesity, uric acid, drinking status, HTN, DM and dyslipidemia. In the Model 2 multivariate logistic regression, only hsCRP levels, uric acid, drinking status, HTN, DM and dyslipidemia remained positively related. According to the results of our study, the hsCRP level was an independent risk factor for MetS. Obesity shares similar risk factors to metabolic syndrome [29], and we also observed a positive relationship between obesity and metabolic syndrome in the univariate logistic regression. The consumption of fructose is higher in the obese population than in the normal population. The metabolism of fructose also elevates serum uric acid [30], and obesity is also linked to metabolic syndrome [31]. We also found a significantly positive relationship between uric acid and metabolic syndrome in both the univariate and multivariate logistic regressions. Elevated plasma glucose levels, triglyceride levels and blood pressure are part of the definition of DM, dyslipidemia and hypertension, respectively [32,33,34]. The definition of metabolic syndrome also includes elevated plasma glucose levels, triglyceride levels and blood pressure [35]. It is not surprising that metabolic syndrome has significant associations with HTN, DM and dyslipidemia in both univariate and multivariate logistic regressions, and this association also corresponded to previous studies [2].

MetS has been considered a risk factor for obesity, HTN, insulin resistance, DM and dyslipidemia, and these diseases associated with MetS have become a heavy burden on health care systems in modern society. A risk factor that can predict MetS would be valuable to alleviate the burden on the health care system related to MetS. People with MetS tend to have more adipocytes, which can produce the chemokines CCL5, monocyte chemotactic protein, macrophage inflammatory protein, macrophage migration inhibition factor and macrophage colony stimulating factor [36,37,38]. These cytokines support the chemotaxis and differentiation of monocytes in adipose tissue, especially in visceral adipose tissue [38]. Eventually, monocytes become macrophages in adipose tissue and secrete proinflammatory cytokines, including IL-6, which stimulate the liver to produce acute-phase reactants, such as hsCRP [4,39].

According to previous cardiovascular disease (CVD) studies, LDL-C in atheromatous plaques can be oxidized and enzyme-modified after the binding of hsCRP. hsCRP can deposit in plaques directly, and the proinflammatory property of hsCRP contributes to the pathogenesis of CVD [40,41] because atherosclerosis is also an inflammatory disease [42]. Furthermore, IL-6, a cytokine that induces hsCRP production in the liver, also accelerates inflammation in atherosclerosis [43]. Overall, the excessive adipocytes in patients with metabolic syndrome elevate proinflammatory substances, including hsCRP and interleukins, in serum. Subsequently, these proinflammatory substances facilitate the atherogenic process from the initial chemotaxis of leukocytes in the arterial wall to the rupture of the plaque [44,45].

The proinflammatory property of hsCRP could also contribute to DM because the inflammation related to CRP alters the endothelial permeability of insulin [46]. The limitation of insulin delivery promotes IR in metabolically active tissue [47].

These studies explained the possible role of hsCRP in the connection between MetS, CVD and IR. Indeed, our study revealed a significant relationship between MetS, CVD and IR. An intensive relationship between hsCRP and MetS was observed even after adjusting for other risk factors. The results of our study not only implied that hsCRP could be an imperative connection of MetS with CVD and IR, but it also indicated that hsCRP could be an independent risk factor for MetS.

Our study had several strengths. First, the study had a sufficient sample size, relevant confounders and appropriate statistics. Second, we confirmed that hsCRP could be an independent risk factor for metabolic syndrome after we considered many related confounders in middle-aged and elderly populations. The results could provide a reference for primary care providers screening for metabolic syndrome. However, there remain some limitations in our study. Regarding the drinking status, although we recorded the frequency of drinking, we did not have the number of types of alcoholic beverages. Since hsCRP levels correlate with alcohol consumption [48], the amount of alcohol intake should be considered in future studies. That all participants came from northern Taiwan is another problem. Selection bias should be considered, so the findings might not represent the entire middle-aged and elderly population.

## 5. Conclusions

In this study, hsCRP was an independent risk factor for MetS in middle-aged and elderly people in northern Taiwan. Thus, our findings could provide valuable information for primary care physicians to alert subjects in this age group regarding the increased risk of MetS.

## Figures and Tables

**Figure 1 ijerph-19-13111-f001:**
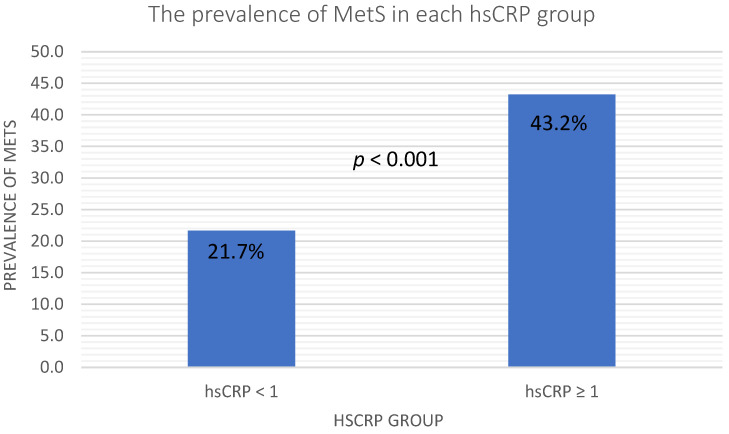
Prevalence of MetS in the different hsCRP groups. Abbreviations: hsCRP, high sensitive C-reactive protein; MetS, metabolic syndrome.

**Table 1 ijerph-19-13111-t001:** General characteristics of participants between non-MetS group and MetS group.

Variables	Total	Non-MetS	MetS	*p* Value
	(n = 400)	(n = 261)	(n = 139)	
Age (year)	64.47 ± 8.45	63.96 ± 8.33	65.42 ± 8.61	0.10
WC (cm)	85.07 ± 9.68	82.00 ± 8.03	90.82 ± 9.92	<0.01
BMI (kg/m^2^)	24.55 ± 3.57	23.44 ± 3.03	26.63 ± 3.58	<0.01
SBP (mmHg)	129.50 ± 16.71	126.11 ± 15.20	135.87 ± 17.59	<0.01
DBP (mmHg)	76.93 ± 11.36	75.89 ± 11.24	78.88 ± 11.38	0.01
Gender, male (%)	141 (35.30%)	94 (36.00%)	47 (33.80%)	0.74
Smoking, n (%)	43 (10.80%)	24 (9.20%)	19 (13.70%)	0.18
Drinking, n (%)	78 (19.50%)	59 (22.60%)	19 (13.70%)	0.03
HTN, n (%)	201 (50.39%)	97 (37.20%)	104 (74.80%)	<0.01
DM, n (%)	79 (19.80%)	25 (9.60%)	54 (38.80%)	<0.01
Dyslipidemia, n (%)	260 (65.00%)	147 (56.30%)	113 (81.30%)	<0.01
hsCRP (mg/L)	1.28 [0.67, 2.42]	1.06 [0.60, 2.07]	1.76 [1.01, 3.84]	<0.01
FPG (mg/dL)	91.00 [83.00, 101.00]	88.00 [82.00, 94.00]	101.00 [88.00, 115.00]	<0.01
HDL-C (mg/dL)	54.43 ± 13.93	58.91 ± 13.11	46.03 ± 11.33	<0.01
TG (mg/dL)	107.00 [77.25, 145.75]	92.00 [71.00, 115.50]	151.00 [113.00, 194.00]	<0.01
LDL-C (mg/dL)	118.37 ± 32.11	118.83 ± 32.12	117.50 ± 32.19	0.69
Total-C (mg/dL)	197.15 ± 35.71	197.31 ± 35.78	196.83 ± 35.69	0.90
Uric Acid (mg/dL)	5.75 ± 1.41	5.57 ± 1.39	6.09 ± 1.38	<0.01

Notes: The categorical variables are expressed as n (%) and were analyzed using the chi-square test. Continuous variables are presented as the mean ± [SD] if the variables were consistent with normally distributed variables and as the median [Q1, Q3] if the variables significantly deviated from a normal distribution. *p* values were obtained from the independent sample *t* test for data consistent with a normal distribution and the Mann-Whitney U test for data consistent with a nonnormal distribution. Abbreviations: WC, waist circumference; BMI, body mass index; SBP, systolic blood pressure; DBP, diastolic blood pressure; HTN, hypertension; DM, diabetes mellitus; hsCRP, high sensitive C-reactive protein; FPG, fasting plasma glucose; HDL-C, high-density lipoprotein cholesterol; TG, triglyceride; LDL-C, low-density lipoprotein cholesterol; Total-C, total cholesterol.

**Table 2 ijerph-19-13111-t002:** The Pearson correlation between cardiometabolic risk factors and hsCRP.

Variables	hs-CRP (n = 400)	
	Pearson’s Coefficient	*p* Value
Age (year)	0.06	0.91
WC (cm)	0.09	0.08
BMI (kg/m2)	0.08	0.13
SBP (mmHg)	0.04	0.45
DBP (mmHg)	−0.01	0.90
FPG (mg/dL)	0.05	0.35
HDL-C (mg/dL)	−0.18	<0.01
TG (mg/dL)	0.09	0.08
LDL-C (mg/dL)	−0.07	0.14
Total-C (mg/dL)	−0.11	0.04
Uric Acid (mg/dL)	0.08	0.13

Abbreviations: hsCRP, high sensitive C-reactive protein; WC, waist circumference; BMI, body mass index; SBP, systolic blood pressure; DBP, diastolic blood pressure; FPG, fasting plasma glucose; HDL-C, high-density lipoprotein cholesterol; TG, triglyceride; LDL-C, low-density lipoprotein cholesterol; Total-C; total cholesterol.

**Table 3 ijerph-19-13111-t003:** Logistic regression analysis of the relationship between cardiometabolic risk factors and MetS.

Variables	Univariate Logistic Regression			Multivariate Logistic Regression		
	OR	(95%CI)	*p* Value	OR	(95%CI)	*p* Value
hsCRP level (≥1 mg/L versus <1 mg/L)	2.75	(1.73–4.36)	<0.01	2.24	(1.23–4.08)	0.01
BMI (Obesity versus non-obesity)	2.40	(1.50–3.85)	<0.01	1.15	(0.61–2.16)	0.67
Uric Acid (mg/dL)	1.30	(1.12–1.51)	<0.01	1.22	(1.00–1.48)	0.05
Age (year)	1.02	(1.00–1.05)	0.10	1.01	(0.98–1.04)	0.48
Gender (men versus women)	1.10	(0.59–1.40)	0.66	1.35	(0.72–2.54)	0.35
Smoking (yes versus no)	1.56	(0.82–2.97)	0.17	1.70	(0.73–3.98)	0.22
Drinking (yes versus no)	0.54	(0.31–0.95)	0.03	0.46	(0.22–0.98)	0.04
HTN (yes versus no)	5.02	(3.18–7.94)	<0.01	4.92	(2.90–8.34)	<0.01
DM (yes versus no)	6.00	(3.51–10.24)	<0.01	5.88	(3.13–11.05)	<0.01
Dyslipidemia (yes versus no)	3.37	(2.06–5.51)	<0.01	3.26	(1.83–5.81)	<0.01

Note: Obesity: BMI ≥ 27 kg/m^2^. Non-obesity: BMI < 27 kg/m^2^. Abbreviations: MetS, metabolic syndrome; hsCRP, high sensitive C-reactive protein; BMI, body mass index; HTN, hypertension; DM, diabetes mellitus.

## Data Availability

The raw data supporting the conclusions of this article will be made available by the corresponding author, without undue reservation.

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
