# Peer review of "The Association between High-Sensitivity C-Reactive Protein and Metabolic Syndrome in an Elderly Population Aged 50 and Older in a Community Receiving Primary Health Care in Taiwan"

_ijerph, 2022, doi:10.3390/ijerph192013111_

Round 1
Reviewer 1 Report
The manuscript "The association between high-sensitive C-reactive protein and 2 metabolic syndrome in an elderly population aged 50 and over 3 in a community of primary health care in Taiwan" focuses on the correlation between hsCRP with metabolic syndrome and other related metabolic risk factors. The results of the study indicate that hs CRP could be considered as an independent factor for metabolic syndrome having multiple benefits for primary care and disease prevention.
Overall, the article needs careful English editing and re-writing of some parts to improve the clarity. Furthermore, the introduction is very short without clarifying the literature gap and the importance of the present study or whether there are other similar studies.
In addition, a graphical abstract is recommended.
There are several typo mistakes throughout the text that must be corrected.
There are several repeated parts that do not follow the standard article structure.
For example "This study recruited 400 participants through posters and notifications from the com- 101 munity office. Three subjects with incomplete data and sixteen subjects with hsCRP 102 >10mg/dl were excluded. The final number of participants was 381." has already been described in the Methods section and should not be repeated in the Results.
Line 149: "non-metabolic group and metabolic group" Please clarify
Limitations of the study need to be added
Author Response
The manuscript "The association between high-sensitive C-reactive protein and 2 metabolic syndrome in an elderly population aged 50 and over 3 in a community of primary health care in Taiwan" focuses on the correlation between hsCRP with metabolic syndrome and other related metabolic risk factors. The results of the study indicate that hs CRP could be considered as an independent factor for metabolic syndrome having multiple benefits for primary care and disease prevention.
Response: 
We thank reviewer for comment
Overall, the article needs careful English editing and re-writing of some parts to improve the clarity. Furthermore, the introduction is very short without clarifying the literature gap and the importance of the present study or whether there are other similar studies.
Response: 
We agreed reviewer’s comprehensive concern, we rewrote our Introduction as below, and whole manuscript underwent English editing
Introduction:
“With wealthy lifestyles and longevity, metabolic syndrome (MetS) has become the most important issue in family medicine and primary care because it is a cluster of metabolic abnormalities, such as overweight, glucose intolerance, elevated blood pressure (BP) and dyslipidemia [1], leading to a higher risk of diabetes mellitus (DM) and cardiovascular disease [2, 3]. Moreover, people with MetS tend to have more adipose tissue, resulting in excessive release of free fatty acids, thereby reducing peripheral insulin sensitivity and causing insulin resistance (IR) [4]. Expanded adipose tissue also leads to the overproduction of proinflammatory cytokines, including C-reactive protein (CRP), interleukin-6 (IL-6), and tumor necrosis factor alpha (TNF-α), by monocyte-derived macrophages in adipose tissue [5, 6]. In addition to IR, inflammation appears to be a pathophysiologic phenomenon of MetS. Several studies have shown that high-sensitivity CRP (hsCRP) is positively associated with fasting insulin, IR and MetS [7][8][9][10]. Similar correlations have been found in different age groups (children and adults) [11][12][13]. Previous studies have also indicated that hsCRP positively correlates with many chronic diseases, such as hypertension (HTN), DM, and dyslipidemia [14][15][16]. Lifestyle can also have an impact on hsCRP levels [17]. Conversely, the prevalence of chronic diseases and metabolic syndrome increases as people age [18][19]. Furthermore, lifestyle factors, such as smoking and drinking, can elevate the hsCRP level [17]. Elevated hsCRP has seemed to be impacted by multiple factors. However, there has been a lack of studies that have considered comprehensive parameters in the relationships between hsCRP and metabolic syndrome. In our research, we gathered many factors, from laboratory data to anthropometric parameters, to evaluate the association between hsCRP levels and metabolic syndrome among elderly individuals aged 50 years old and older in northern Taiwan. The results of our research could provide a reference for primary care providers to survey metabolic syndrome. “ (line:27-51)
Reference:
- Heianza, Y., Kato, K., Kodama, S., Ohara, N., Suzuki, A., Tanaka, S., ... & Sone, H. (2015). Risk of the development of Type 2 diabetes in relation to overall obesity, abdominal obesity and the clustering of metabolic abnormalities in Japanese individuals: does metabolically healthy overweight really exist? The Niigata Wellness Study. Diabetic Medicine, 32(5), 665-672.
- Grundy, S.M., MetS: connecting and reconciling cardiovascular and diabetes worlds. Journal of the American College of Cardiology, 2006. 47(6): p. 1093-100.
- Sutherland, J.P., B. McKinley, and R.H. Eckel, The MetS and inflammation. MetS and related disorders, 2004. 2(2): p. 82-104.
- Russo, L., & Lumeng, C. N. (2018). Properties and functions of adipose tissue macrophages in obesity. Immunology, 155(4), 407-417.
- Matulewicz, N., & Karczewska-Kupczewska, M. (2016). Insulin resistance and chronic inflammation. Postepy higieny i medycyny doswiadczalnej (Online), 70, 1245-1258.
- Shrivastava, A. K., Singh, H. V., Raizada, A., & Singh, S. K. (2015). C-reactive protein, inflammation and coronary heart disease. The Egyptian Heart Journal, 67(2), 89-97.
- Drabsch, T., Holzapfel, C., Stecher, L., Petzold, J., Skurk, T., & Hauner, H. (2018). Associations between C-reactive protein, insulin sensitivity, and resting metabolic rate in adults: A mediator analysis. Frontiers in endocrinology, 9, 556.
- Chen, L., Chen, R., Wang, H., & Liang, F. (2015). Mechanisms linking inflammation to insulin resistance. International journal of endocrinology, 2015.
- Xu, W., Tian, M., & Zhou, Y. (2018). The relationship between insulin resistance, adiponectin and C-reactive protein and vascular endothelial injury in diabetic patients with coronary heart disease. Experimental and therapeutic medicine, 16(3), 2022-2026.
- Yousuf, O., Mohanty, B. D., Martin, S. S., Joshi, P. H., Blaha, M. J., Nasir, K., ... & Budoff, M. J. (2013). High-sensitivity C-reactive protein and cardiovascular disease: a resolute belief or an elusive link?. Journal of the American College of Cardiology, 62(5), 397-408.
- Kyithar, M. P., Bonner, C., Bacon, S., Kilbride, S. M., Schmid, J., Graf, R., ... & Byrne, M. M. (2013). Effects of hepatocyte nuclear factor-1A and-4A on pancreatic stone protein/regenerating protein and C-reactive protein gene expression: implications for maturity-onset diabetes of the young. Journal of translational medicine, 11(1), 1-12.
- Suhett, L. G., Hermsdorff, H. H. M., Rocha, N. P., Silva, M. A., Filgueiras, M. D. S., Milagres, L. C., ... & Novaes, J. F. D. (2019). Increased C-reactive protein in Brazilian children: association with cardiometabolic risk and metabolic syndrome components (PASE study). Cardiology Research and Practice, 2019.
- Josune Olza, Concepción M.A., Mercedes Gil-Campos, Rosaura Leis, Gloria Bueno Miguel Valle Ramón Cañete, Rafael Tojo, Luis A. Moreno, Ángel Gil, A Continuous MetS Score Is Associated with Specific Biomarkers of Inflammation and CVD Risk in Prepubertal Children Ann Nutr Metab 2015;66:72–79.
- Lakoski SG, Cushman M, Siscovick DS, et al. The relationship between inflammation, obesity and risk for hypertension in the Muti-Ethnic Study of Atherosclerosis (MESA) J Hum Hypertens. 2011;25:73–79.
- Noordam R, Oudt CH, Bos MM, Smit RAJ, van Heemst D. High-sensitivity C-reactive protein, low-grade systemic inflammation and type 2 diabetes mellitus: A two-sample Mendelian randomization study. Nutr Metab Cardiovasc Dis. 2018 Aug;28(8):795-802.
- Koenig W. Low-Grade Inflammation Modifies Cardiovascular Risk Even at Very Low LDL-C Levels: Are We Aiming for a Dual Target Concept? Circulation. 2018 Jul 10;138(2):150-153. Koenig W. Low-Grade Inflammation Modifies Cardiovascular Risk Even at Very Low LDL-C Levels: Are We Aiming for a Dual Target Concept? Circulation. 2018 Jul 10;138(2):150-153.
- Blaum C, Brunner FJ, Kröger F, Braetz J, Lorenz T, Goßling A, Ojeda F, Koester L, Karakas M, Zeller T, Westermann D, Schnabel R, Blankenberg S, Seiffert M, Waldeyer C. Modifiable lifestyle risk factors and C-reactive protein in patients with coronary artery disease: Implications for an anti-inflammatory treatment target population. Eur J Prev Cardiol. 2021 Apr 10;28(2):152–158.
In addition, a graphical abstract is recommended.
Response: 
We thank reviewer for reminding. We added Figure 1 as graphical abstract to demonstrate the prevalence of metabolic syndrome in each group.
There are several typo mistakes throughout the text that must be corrected.
Response: 
We appreciate the reviewer’s comments. We checked our manuscript, and the English editing has been done again.
There are several repeated parts that do not follow the standard article structure. For example "This study recruited 400 participants through posters and notifications from the com- 101 munity office. Three subjects with incomplete data and sixteen subjects with hsCRP 102 >10mg/dl were excluded. The final number of participants was 381." has already been described in the Methods section and should not be repeated in the Results.
Response: 
Thanks for the reviewer’s suggestion. We modified our article and removed repeat content.
Line 149: "non-metabolic group and metabolic group" Please clarify
Response: 
We thank the reviewer for the comments. We modified our content in Material and Method as below:
“MetS was defined according to the modified criteria of the National Cholesterol Education Program Adult Treatment Panel III (NCEP-ATP III) [19], but the standard of WC has been adjusted by the Minister of Health and Welfare of Taiwan. MetS is defined by the presence of three or more of the following components: (1) WC ≥ 90 cm for men and ≥ 80 cm for women; (2) TG ≥150 mg/dL; (3) HDL-C < 40 mg/dL for men and < 50 mg/dL for women; (4) BP ≥ 130/85 mm Hg or current use of antihypertensive medications; and (5) FPG ≥ 100 mg/dL. The participants who met the criteria of MetS were classified in metabolic group. Whereas, the participants who did not meet the criteria were classified in non-metabolic group.” (line:86-94)
Reference:
- Kassi, E., Pervanidou, P., Kaltsas, G., & Chrousos, G. (2011). Metabolic syndrome: definitions and controversies. BMC medicine, 9(1), 1-13.
Limitations of the study need to be added
Response: 
We agree with the reviewer regarding the fact that several limitation should be concerned in our article. We add limitation in Discussion as below:
“However, there remain some limitations in our study. Regarding drinking status, although we recorded the frequency of drinking, we did not have the number of type of alcoholic beverages. Since hsCRP levels correlate with alcohol consumption,[50] the amount of alcohol intake should be considered in future studies. That all participants came from Northern Taiwan is another problem. Selection bias should be considered, so the findings might not represent the entire middle-aged and elderly population.” (line: 255-261)
Reference :
- Park, J. Y., Kim, M. J., & Kim, J. H. (2019). Influence of Alcohol Consumption on the Serum hs-CRP Level and Prevalence of Metabolic Syndrome-Based on the 2015 Korean National Health and Nutrition Examination Survey. Journal of the Korean Dietetic Association, 25(2), 83-104.

Reviewer 2 Report
The manuscript is interesting, well written and points out relevant issues.
My comments:
· Line 13: after “adipose tissue,” must be a period instead of a comma.
· I suggest that authors introduce some kind of study conclusion at the end of the abstract.
· Line 26-27: in the text appears “and primary care, Because metabolic syndrome”. Suggestion: “and primary care. Metabolic syndrome”.
· Line 29: in the text appears “and dyslipidemia, [1] leading”. Suggestion: “and dyslipidemia [1], leading”.
· Line 30: in the text appears “and cardiovascular disease. [2, 3] Moreover, People”. Suggestion: “and cardiovascular disease [2, 3]. Moreover, people”. The same suggestion to change the punctuation after placing the references must be considered throughout the document.
· Line 41: in the text appears “with other rism,k factors among”. Suggestion: “with other risk factors among”.
· Line 126: in the text appears “HDL”. Suggestion: “HDL-C”. To maintain consistency.
· Lines 127: in the text appears “and Total-C are not significant”. Suggestion “and Total-C were not significant”.
· Lines 159-160: in the text appears “hsCRP had positive associations with WC, SBP, DBP, FPG, and TG.”. Suggestion “hsCRP had positive associations with WC, SBP, FPG and TG.”. Authors should confirm this.
· Lines 165: improve the sentence “Table 3 was logistic regression analysis.”
· Lines 167-168: improve the sentence “In model 2 multivariate logistic regression, only hsCRP, uric acid, HTN, DM, and dyslipidemia remained positive relationship.”. Maybe something like “In model 2 (multivariate logistic regression), only hsCRP, uric acid, HTN, DM, and dyslipidemia remained with a positive relationship.”.
As an additional opinion: In the conclusions of the study, the authors suggest that hsCRP may be a risk factor for MetS. However, it is important to be aware that pathophysiologically hsCRP does not seem to motivate MetS, but CVD. MetS is also an indicator of increased CV risk. So, it is to be expected that these may be correlated.
Considering that individuals with MetS have, mostly, excess body fat, mainly in the abdominal region, this may motivate the greater production of hsCRP by adipocytes.
The authors should mention the limitations of the study.
Congratulations on the study!
Author Response
The manuscript is interesting, well written and points out relevant issues.
Response: 
We thank reviewer for comment
My comments:
- Line 13: after “adipose tissue,” must be a period instead of a comma.
Response: 
We thank the reviewer for the notification. The manuscript underwent English editing after we revised the content.
- I suggest that authors introduce some kind of study conclusion at the end of the abstract.
Response: 
We thank the reviewer for the valuable comments. We add study conclusion in Abstract as below:
“In summary, our research indicated that hsCRP could be an independent risk factor for MetS.”
(Line:21-22)
- Line 26-27: in the text appears “and primary care, Because metabolic syndrome”. Suggestion: “and primary care. Metabolic syndrome”.
Response: 
We thank the reviewer for the notification. The manuscript underwent English editing after we revised the content.
- Line 29: in the text appears “and dyslipidemia, [1] leading”. Suggestion: “and dyslipidemia [1], leading”.
Response: 
We thank the reviewer for the notification. The manuscript underwent English editing after we revised the content.
- Line 30: in the text appears “and cardiovascular disease. [2, 3] Moreover, People”. Suggestion: “and cardiovascular disease [2, 3]. Moreover, people”. The same suggestion to change the punctuation after placing the references must be considered throughout the document.
Response: 
We thank the reviewer for the notification. The manuscript underwent English editing after we revised the content.
- Line 41: in the text appears “with other rism,k factors among”. Suggestion: “with other risk factors among”.
Response: 
We thank the reviewer for the notification. The manuscript underwent English editing after we revised the content.
- Line 126: in the text appears “HDL”. Suggestion: “HDL-C”. To maintain consistency.
Response: 
We thank the reviewer for the notification. The manuscript underwent English editing after we revised the content.
- Lines 127: in the text appears “and Total-C are not significant”. Suggestion “and Total-C were not significant”.
Response: 
We thank the reviewer for the notification. The manuscript underwent English editing after we revised the content.
- Lines 159-160: in the text appears “hsCRP had positive associations with WC, SBP, DBP, FPG, and TG.”. Suggestion “hsCRP had positive associations with WC, SBP, FPG and TG.”. Authors should confirm this.
Response: 
We thank the reviewer for the notification. The manuscript underwent English editing after we revised the content.
- Lines 165: improve the sentence “Table 3 was logistic regression analysis.”
Response: 
We thank the reviewer for the notification. The manuscript underwent English editing after we revised the content.
- Lines 167-168: improve the sentence “In model 2 multivariate logistic regression, only hsCRP, uric acid, HTN, DM, and dyslipidemia remained positive relationship.”. Maybe something like “In model 2 (multivariate logistic regression), only hsCRP, uric acid, HTN, DM, and dyslipidemia remained with a positive relationship.”.
Response: 
We thank the reviewer for the notification. The manuscript underwent English editing after we revised the content.
As an additional opinion: In the conclusions of the study, the authors suggest that hsCRP may be a risk factor for MetS. However, it is important to be aware that pathophysiologically hsCRP does not seem to motivate MetS, but CVD. MetS is also an indicator of increased CV risk. So, it is to be expected that these may be correlated.Considering that individuals with MetS have, mostly, excess body fat, mainly in the abdominal region, this may motivate the greater production of hsCRP by adipocytes.
Response: 
We agreed reviewer’s comprehensive concern. We do some literature review and rewrote the related content in discussion as below:
“According to previous cardiovascular disease (CVD) studies, LDL-C in atheroma-tous plaques can be oxidized and enzyme modified after the binding of hsCRP. hsCRP can deposit in plaques directly, and the proinflammatory property of hsCRP contributes to the pathogenesis of CVD [42][43] because atherosclerosis is also an inflammatory disease [44]. Furthermore, IL-6, a cytokine that induces hsCRP production in the liver, also accelerates inflammation in atherosclerosis [45]. Overall, the excessive adipocytes in patients with metabolic syndrome elevate proinflammatory substances, including hsCRP and interleukins, in serum. Subsequently, these proinflammatory substances facilitate the atherogenic process from the initial chemotaxis of leukocytes [46] in the arterial wall to the rupture of the plaque [47]”(Line: 232-241)
Reference:
- Denegri, A., & Boriani, G. (2021). High sensitivity C-reactive protein (hsCRP) and its implications in cardiovascular outcomes. Current Pharmaceutical Design, 27(2), 263-275.
- Singh, U., Dasu, M. R., Yancey, P. G., Afify, A., Devaraj, S., & Jialal, I. (2008). Human C-reactive protein promotes oxidized low density lipoprotein uptake and matrix metalloproteinase-9 release in Wistar rats. Journal of lipid research, 49(5), 1015-1023.
- Tuttolomondo, A., Di Raimondo, D., Pecoraro, R., Arnao, V., Pinto, A., & Licata, G. (2012). Atherosclerosis as an inflammatory disease. Current pharmaceutical design, 18(28), 4266-4288.
- Tyrrell, D. J., & Goldstein, D. R. (2021). Ageing and atherosclerosis: vascular intrinsic and extrinsic factors and potential role of IL-6. Nature Reviews Cardiology, 18(1), 58-68.
- Soeki, T., & Sata, M. (2016). Inflammatory biomarkers and atherosclerosis. International heart journal, 15-346.
- Rogoveanu, O. C., MogoÅŸanu, G. D., Bejenaru, C., Bejenaru, L. E., Croitoru, O., NeamÅ£u, J., ... & Scorei, R. I. (2015). Effects of calcium fructoborate on levels of C-reactive protein, total cholesterol, low-density lipoprotein, triglycerides, IL-1β, IL-6, and MCP-1: a double-blind, placebo-controlled clinical study. Biological trace element research, 163(1), 124-131.
The authors should mention the limitations of the study.
Response: 
We agree with the reviewer regarding the fact that several limitation should be concerned in our article. We add limitation in Discussion as below:
“However, there remain some limitations in our study. Regarding drinking status, although we recorded the frequency of drinking, we did not have the number of type of alcoholic beverages. Since hsCRP levels correlate with alcohol consumption,[50] the amount of alcohol intake should be considered in future studies. That all participants came from Northern Taiwan is another problem. Selection bias should be considered, so the findings might not represent the entire middle-aged and elderly population.” (Line:255-261)
Reference :
- Park, J. Y., Kim, M. J., & Kim, J. H. (2019). Influence of Alcohol Consumption on the Serum hs-CRP Level and Prevalence of Metabolic Syndrome-Based on the 2015 Korean National Health and Nutrition Examination Survey. Journal of the Korean Dietetic Association, 25(2), 83-104.
Congratulations on the study!
Response: 
We thank the reviewer for the comment

Reviewer 3 Report
It's a well-written study. However, the following supplements are needed:
The association of hsCRP with metabolic syndrome and its components is a well-known study. It is necessary to demonstrate that the study complements the limitations of the study rather than reconfirming the results of well-known existing studies. Since the subjects of the study were 50 years of age or older, the characteristics of the elderly and middle-aged were at the same time. It is also well known that the prevalence of metabolic syndrome increases with age. It is necessary to add research findings that complement the limitations of existing studies.
Line 41: rism,k factors âžž risk factors
Line 56 to 59: Chronic diseases associated with elevated hsCRP should also be ruled out. This needs to be evaluated.
Line 109 to 110: In general, hsCRP does not perform a normal distribution. Have you verified that this data has a normal distribution? If so, please mention the results.
Line 115: Alcohol consumption was not analyzed by the amount of alcohol consumed, but by the presence or absence of alcohol consumption. Since hsCRP has a positive correlation with the amount of alcohol consumed, a comparative analysis is required with the amount of alcohol consumed.
Line 124 to 127: If Spearman’s correlation is less than 0.2, it is evaluated as having little correlation regardless of statistical significance. More useful results may be obtained by dividing hsCRP by a baseline (1.0 mg/dL) or higher baseline and then analyzing the OR.
Lines 138 to 144:
Since the risk of metabolic syndrome was assessed with hsCRP as a continuous variable, the criteria for OR's comparison are not known. Please compare with the standard of 1.0 mg/dL or higher, which is the normal standard of the Center for Disease Control and the American Heart Association.
Although we were comparing the risks for the components of the metabolic syndrome, Table 3 shows only some of the components of the metabolic syndrome. After classifying hsCRP as normal or elevated levels, it is necessary to analyze the OR of each of the components of the metabolic syndrome.
Table The abbreviations below and definitions of HTN, DM, and Dyslipidemia are required. HTN, DM, and Dyslipidemia need not be explained in the case of replacement with a component of metabolic syndrome.
Author Response
It's a well-written study. However, the following supplements are needed:
The association of hsCRP with metabolic syndrome and its components is a well-known study. It is necessary to demonstrate that the study complements the limitations of the study rather than reconfirming the results of well-known existing studies. Since the subjects of the study were 50 years of age or older, the characteristics of the elderly and middle-aged were at the same time. It is also well known that the prevalence of metabolic syndrome increases with age. It is necessary to add research findings that complement the limitations of existing studies.
Response: 
We agreed reviewer’s comprehensive concern. We do some literature review and added content in the Abstract as below:
“Previous studies have also indicated that hsCRP positively correlates with many chronic diseases, such as hypertension (HTN), DM, and dyslipidemia [14][15][16]. Lifestyle can also have an impact on hsCRP levels [17]. Conversely, the prevalence of chronic diseases and metabolic syndrome increases as people age [18]. Furthermore, lifestyle factors, such as smoking and drinking, can elevate the hsCRP level [17]. Elevated hsCRP has seemed to be impacted by multiple factors. However, there has been a lack of studies that have considered comprehensive parameters in the relationships between hsCRP and metabolic syndrome. In our research, we gathered many factors, from laboratory data to anthropometric parameters, to evaluate the association between hsCRP levels and metabolic syndrome among elderly individuals aged 50 years old and older in northern Taiwan. The results of our research could provide a reference for primary care providers to survey metabolic syndrome.” (Line:38-49)
Reference:
- Lakoski SG, Cushman M, Siscovick DS, et al. The relationship between inflammation, obesity and risk for hypertension in the Muti-Ethnic Study of Atherosclerosis (MESA) J Hum Hypertens. 2011;25:73–79.
- Noordam R, Oudt CH, Bos MM, Smit RAJ, van Heemst D. High-sensitivity C-reactive protein, low-grade systemic inflammation and type 2 diabetes mellitus: A two-sample Mendelian randomization study. Nutr Metab Cardiovasc Dis. 2018 Aug;28(8):795-802.
- Koenig W. Low-Grade Inflammation Modifies Cardiovascular Risk Even at Very Low LDL-C Levels: Are We Aiming for a Dual Target Concept? Circulation. 2018 Jul 10;138(2):150-153. Koenig W. Low-Grade Inflammation Modifies Cardiovascular Risk Even at Very Low LDL-C Levels: Are We Aiming for a Dual Target Concept? Circulation. 2018 Jul 10;138(2):150-153.
- Blaum C, Brunner FJ, Kröger F, Braetz J, Lorenz T, Goßling A, Ojeda F, Koester L, Karakas M, Zeller T, Westermann D, Schnabel R, Blankenberg S, Seiffert M, Waldeyer C. Modifiable lifestyle risk factors and C-reactive protein in patients with coronary artery disease: Implications for an anti-inflammatory treatment target population. Eur J Prev Cardiol. 2021 Apr 10;28(2):152–158.
- Liu B, Chen G, Zhao R, Huang D, Tao L. Temporal trends in the prevalence of metabolic syndrome among middle-aged and elderly adults from 2011 to 2015 in China: the China health and retirement longitudinal study (CHARLS). BMC Public Health. 2021 Jun 2;21(1):1045
Line 41: rism,k factors âžž risk factors
Response: 
We thank the reviewer for the notification. The manuscript underwent English editing after we revised the content.
Line 56 to 59: Chronic diseases associated with elevated hsCRP should also be ruled out. This needs to be evaluated.
Response: 
We thank the reviewer’s concern, and we included all the participants in the new database. We renewed all the tables, figure and content in our manuscript, based on the new database.
Line 109 to 110: In general, hsCRP does not perform a normal distribution. Have you verified that this data has a normal distribution? If so, please mention the results.
Response: 
We totally agreed with the reviewer’s comment. The normality of the continuous variables was checked by the Shapiro-Wilk normality test. Continuous variables are presented as the mean ± [SD] if the variables were consistent with normally distributed variables and as the median [Q1, Q3] if the variables significantly deviated from a normal distribution. We obtained p values from the independent sample t test for data consistent with a normal distribution and the Mann-Whitney U test for data consistent with a non-normal distribution. The related content was modified, and we add the description in Material and Method as below:
“In Table 1, the categorical variables are expressed as n (%) and were analyzed using the chi-square test. The normality of the continuous variables was checked by the Shapiro-Wilk normality test. Continuous variables are presented as the mean ± [SD] if the variables (age, WC, BMI, SBP, DBP, HDL-C, LDL-C, total C, and uric acid) were consistent with normally distributed variables and as the median [Q1, Q3] if the variables (hsCRP, FPG, and TG) significantly deviated from a normal distribution. We obtained p values from the independent sample t test for data consistent with a normal distribution and the Mann-Whitney U test for data consistent with a nonnormal distribution.” (Line:104 -112)
Line 115: Alcohol consumption was not analyzed by the amount of alcohol consumed, but by the presence or absence of alcohol consumption. Since hsCRP has a positive correlation with the amount of alcohol consumed, a comparative analysis is required with the amount of alcohol consumed.
Response: 
We appreciate the reviewer’s concern. However, our questionnaire did not survey the amount of alcohol consumption. We will include the amount of alcohol consumption in our future studies, and we added your concern to Limitation as below:
“However, there remain some limitations in our study. Regarding drinking status, although we recorded the frequency of drinking, we did not have the number of type of alcoholic beverages. Since hsCRP levels correlate with alcohol consumption,[50] the amount of alcohol intake should be considered in future studies.”(Line: 255-259)
Reference:
- Park, J. Y., Kim, M. J., & Kim, J. H. (2019). Influence of Alcohol Consumption on the Serum hs-CRP Level and Prevalence of Metabolic Syndrome-Based on the 2015 Korean National Health and Nutrition Examination Survey. Journal of the Korean Dietetic Association, 25(2), 83-104
Line 124 to 127: If Spearman’s correlation is less than 0.2, it is evaluated as having little correlation regardless of statistical significance. More useful results may be obtained by dividing hsCRP by a baseline (1.0 mg/dL) or higher baseline and then analyzing the OR.
Response: 
We thank you for the suggestion. We defined elevated hsCRP as hsCRP ≥ 1 mg/L, and we used this new classification in Figure 1 and Table 3. The related content was also adjusted as below:
“Based on the Centers for Disease Control and the American Heart Association, elevated hsCRP was defined as plasma hsCRP ≥ 1 mg/L for higher risk of cardiovascular disease [21] [22].” (Line: 100-102)
“In Figure 1, the elevated hsCRP was defined as hsCRP ≥ 1 mg/L, and we observed a higher prevalence of MetS in the elevated hsCRP group. The prevalence of MetS in the normal hsCRP group and elevated hsCRP group were 21.7 and 43.2 respectively with a p-value < 0.001.” (Line:162-165)
“elevated hsCRP was defined as hsCRP ≥ 1 mg/L.” (Line:169)
Reference:
- Sabatine, M. S., Morrow, D. A., Jablonski, K. A., Rice, M. M., Warnica, J. W., Domanski, M. J., ... & Braunwald, E. (2007). Prognostic significance of the Centers for Disease Control/American Heart Association high-sensitivity C-reactive protein cut points for cardiovascular and other outcomes in patients with stable coronary artery disease. Circulation, 115(12), 1528-1536.
- Nehring, S. M., Goyal, A., Bansal, P., & Patel, B. C. (2017). C reactive protein.
Lines 138 to 144:
Since the risk of metabolic syndrome was assessed with hsCRP as a continuous variable, the criteria for OR's comparison are not known. Please compare with the standard of 1.0 mg/dL or higher, which is the normal standard of the Center for Disease Control and the American Heart Association.
Response: 
We thank the reviewer for the valuable comments. We defined elevated hsCRP as hsCRP ≥ 1 mg/L, according to the Centers for Disease Control and the American Heart Association. The data was also analyzed again based on this new classification in Figure 1 and Table 3.
Although we were comparing the risks for the components of the metabolic syndrome, Table 3 shows only some of the components of the metabolic syndrome. After classifying hsCRP as normal or elevated levels, it is necessary to analyze the OR of each of the components of the metabolic syndrome.
Response: 
We appreciated the reviewer’s constructive comments. We defined elevated hsCRP as hsCRP ≥ 1 mg/L, and the data in Table 3 was also analyzed again under this new classification.
Table The abbreviations below and definitions of HTN, DM, and Dyslipidemia are required. HTN, DM, and Dyslipidemia need not be explained in the case of replacement with a component of metabolic syndrome.
Response:
We agree with the reviewer regarding the abbreviation and explanations. We removed redundant explanations and added the definition of abbreviation under every table and figure

Reviewer 4 Report
Major Comments:
1. Although WC is included as one of the parameter to define MetS, it would be also appropriate if author can report BMI status of the study participants, and see how the variables and logistic regression behave after adjusting for BMI.
2. Age seems to be significantly different (P=0.01) between MetS and Non MetS group? So it would be appropriate if authors can report age adjusted parameters. Also, more appropriate would be to re-analyze the results logistic regression (table 3 )and evaluate the relation between cardio-metabolic risk factors and MetS after adjusting for gender, smoking and drinking status of participants.
Other comments:
1. Many typographical errors existed. Extensive revision of introduction, results and discussion is required. Introduction should focus on the significance and novelty of the study.
2. Discussion is too short even for a clinical report.
3. Most of the citations were very old and outdated which is significantly reducing the originality of the present work.
Author Response
Major Comments:
- Although WC is included as one of the parameter to define MetS, it would be also appropriate if author can report BMI status of the study participants, and see how the variables and logistic regression behave after adjusting for BMI.
Response:
We thank the reviewer for the valuable comments. We added BMI data and performed analysis again in all tables. BMI was significantly higher in the metabolic syndrome group in Table 1, but BMI did not show significant relationship in Pearson’s correlation and multivariate logistic regression. The related content was modified as well.
- Age seems to be significantly different (P=0.01) between MetS and Non MetS group? So it would be appropriate if authors can report age adjusted parameters. Also, more appropriate would be to re-analyze the results logistic regression (table 3 )and evaluate the relation between cardio-metabolic risk factors and MetS after adjusting for gender, smoking and drinking status of participants.
Response:
We agreed with the reviewer’s comment. However, age did not show significant relationship in all tables after we modified the analysis which included BMI and the demand from other reviewers. Hence, there was no reason to make adjustments for age anymore. In Table 3, the multivariate logistic regression was adjusted with BMI, uric acid, age, gender, smoking, drinking, HTN, DM, and dyslipidemia. The hsCRP (p = 0.01, OR:2.24, 95% CI:1.23~4.08) still showed significant association. The related content was also revised
Other comments:
- Many typographical errors existed. Extensive revision of introduction, results and discussion is required. Introduction should focus on the significance and novelty of the study.
Response:
We agreed reviewer’s comprehensive concern. The revised manuscript underwent English editing. We done some literature review and added content in the Introduction as below:
“Previous studies have also indicated that hsCRP positively correlates with many chronic diseases, such as hypertension (HTN), DM, and dyslipidemia [14][15][16]. Lifestyle can also have an impact on hsCRP levels [17]. Conversely, the prevalence of chronic diseases and metabolic syndrome increases as people age [18]. Furthermore, lifestyle factors, such as smoking and drinking, can elevate the hsCRP level [17]. Elevated hsCRP has seemed to be impacted by multiple factors. However, there has been a lack of studies that have considered comprehensive parameters in the relationships between hsCRP and metabolic syndrome. In our research, we gathered many factors, from laboratory data to anthropometric parameters, to evaluate the association between hsCRP levels and metabolic syndrome among elderly individuals aged 50 years old and older in northern Taiwan. The results of our research could provide a reference for primary care providers to survey metabolic syndrome.” (Line: 38-49)
Reference:
- Lakoski SG, Cushman M, Siscovick DS, et al. The relationship between inflammation, obesity and risk for hypertension in the Muti-Ethnic Study of Atherosclerosis (MESA) J Hum Hypertens. 2011;25:73–79.
- Noordam R, Oudt CH, Bos MM, Smit RAJ, van Heemst D. High-sensitivity C-reactive protein, low-grade systemic inflammation and type 2 diabetes mellitus: A two-sample Mendelian randomization study. Nutr Metab Cardiovasc Dis. 2018 Aug;28(8):795-802.
- Koenig W. Low-Grade Inflammation Modifies Cardiovascular Risk Even at Very Low LDL-C Levels: Are We Aiming for a Dual Target Concept? Circulation. 2018 Jul 10;138(2):150-153. Koenig W. Low-Grade Inflammation Modifies Cardiovascular Risk Even at Very Low LDL-C Levels: Are We Aiming for a Dual Target Concept? Circulation. 2018 Jul 10;138(2):150-153.
- Blaum C, Brunner FJ, Kröger F, Braetz J, Lorenz T, Goßling A, Ojeda F, Koester L, Karakas M, Zeller T, Westermann D, Schnabel R, Blankenberg S, Seiffert M, Waldeyer C. Modifiable lifestyle risk factors and C-reactive protein in patients with coronary artery disease: Implications for an anti-inflammatory treatment target population. Eur J Prev Cardiol. 2021 Apr 10;28(2):152–158.
- Liu B, Chen G, Zhao R, Huang D, Tao L. Temporal trends in the prevalence of metabolic syndrome among middle-aged and elderly adults from 2011 to 2015 in China: the China health and retirement longitudinal study (CHARLS). BMC Public Health. 2021 Jun 2;21(1):1045
- Discussion is too short even for a clinical report.
Response:
We totally agreed with the reviewer’s concern. We have done extensive study and enriched our discussion as below:
“The consumption of fructose is higher in the obese population than in the normal population. The metabolism of fructose also elevates serum uric acid [30], and obesity is also linked to metabolic syndrome [31]. We also found a significantly positive relationship between uric acid and metabolic syndrome in both univariate and multivariate logistic regressions. Elevated plasma glucose levels, triglyceride levels, and blood pressure are part of definition of DM, dyslipidemia, and hypertension, respectively [32][33][34]. The definition of metabolic syndrome also includes elevated plasma glucose levels, triglyceride levels and blood pressure [35]. It is not surprising that metabolic syndrome has significant associations with HTN, DM, and dyslipidemia in both univariate and multivariate logistic regressions, and this association also corresponded to previous studies [2].” (Line:211-220)
“because atherosclerosis is also an inflammatory disease [44]. Furthermore, IL-6, a cytokine that induces hsCRP production in the liver, also accelerates inflammation in atherosclerosis [45]. Overall, the excessive adipocytes in patients with metabolic syndrome elevate proinflammatory substances, including hsCRP and interleukins, in serum. Subsequently, these proinflammatory substances facilitate the atherogenic process from the initial chemotaxis of leukocytes [46] in the arterial wall to the rupture of the plaque [47].” (Line:235-241)
“Our study had several strengths. First, the study had a sufficient sample size, rele-vant confounders, and appropriate statistics. Second, we confirmed that hsCRP could be an independent risk factor for metabolic syndrome after we considered many related confounders in middle-aged and elderly populations. The results could provide a reference for primary care providers to screen for metabolic syndrome. However, there remain some limitations in our study. Regarding drinking status, although we recorded the frequency of drinking, we did not have the number of type of alcoholic beverages. Since hsCRP levels correlate with alcohol consumption,[50] the amount of alcohol intake should be considered in future studies. That all participants came from Northern Taiwan is another problem. Selection bias should be considered, so the findings might not represent the entire middle-aged and elderly population.” (Line:251-261)
Reference:
- Grundy, S.M., MetS: connecting and reconciling cardiovascular and diabetes worlds. Journal of the American College of Cardiology, 2006. 47(6): p. 1093-100.
- Caliceti C, Calabria D, Roda A, Cicero AFG. Fructose Intake, Serum Uric Acid, and Cardiometabolic Disorders: A Critical Review. Nutrients. 2017 Apr 18;9(4):395.
- Han, T. S., & Lean, M. E. (2016). A clinical perspective of obesity, metabolic syndrome and cardiovascular disease. JRSM cardiovascular disease, 5, 2048004016633371.
- Inzucchi, S. E. (2012). Diagnosis of diabetes. New England Journal of Medicine, 367(6), 542-550.
- Wu, J. Y., Duan, X. Y., Li, L., Dai, F., Li, Y. Y., Li, X. J., & Fan, J. G. (2010). Dyslipidemia in shanghai, China. Preventive medicine, 51(5), 412-415.
- Parati, G., Stergiou, G., O’Brien, E., Asmar, R., Beilin, L., Bilo, G., ... & Zhang, Y. (2014). European Society of Hypertension practice guidelines for ambulatory blood pressure monitoring. Journal of hypertension, 32(7), 1359-1366.
- Parikh, R. M., & Mohan, V. (2012). Changing definitions of metabolic syndrome. Indian journal of endocrinology and metabolism, 16(1), 7.
- Tuttolomondo, A., Di Raimondo, D., Pecoraro, R., Arnao, V., Pinto, A., & Licata, G. (2012). Atherosclerosis as an inflammatory disease. Current pharmaceutical design, 18(28), 4266-4288.
- Tyrrell, D. J., & Goldstein, D. R. (2021). Ageing and atherosclerosis: vascular intrinsic and extrinsic factors and potential role of IL-6. Nature Reviews Cardiology, 18(1), 58-68.
- Soeki, T., & Sata, M. (2016). Inflammatory biomarkers and atherosclerosis. International heart journal, 15-346.
- Rogoveanu, O. C., MogoÅŸanu, G. D., Bejenaru, C., Bejenaru, L. E., Croitoru, O., NeamÅ£u, J., ... & Scorei, R. I. (2015). Effects of calcium fructoborate on levels of C-reactive protein, total cholesterol, low-density lipoprotein, triglycerides, IL-1β, IL-6, and MCP-1: a double-blind, placebo-controlled clinical study. Biological trace element research, 163(1), 124-131.
- Park, J. Y., Kim, M. J., & Kim, J. H. (2019). Influence of Alcohol Consumption on the Serum hs-CRP Level and Prevalence of Metabolic Syndrome-Based on the 2015 Korean National Health and Nutrition Examination Survey. Journal of the Korean Dietetic Association, 25(2), 83-104.
- Most of the citations were very old and outdated which is significantly reducing the originality of the present work.
Response:
We appreciate the reviewer’s comments, and the outdated reference had been replaced in revised manuscript.

Round 2
Reviewer 3 Report
Thank you. Please correct a few things below.
1. Please correct the contents that do not comply with the method of notation of references.
2. After the conclusion, please add the following items in compliance with the submission rules.
Institutional Review Board Statement:
Informed Consent Statement:
Data Availability Statement:
Author Response
- Please correct the contents that do not comply with the method of notation of references.
Ans:
We appreciated your notification, and we corrected all the citations to comply with the notation of reference.
- After the conclusion, please add the following items in compliance with the submission rules.
Ans:
We thank reviewer for the advice. The statements were added at the end of the article as below:
Institutional Review Board Statement:
The studies involving human participants were reviewed and approved by Chang Gung Medical Foundation Institutional Review Board (102-2304B).
Informed Consent Statement:
We explained the informed consent to the patients/participants, and they provided their written informed consent before they participated in this study.
Data Availability Statement:
The raw data supporting the conclusions of this article will be made available by the corresponding author, without undue reservation.

Reviewer 4 Report
No further suggestions.
Author Response
Thank you for your comments, which improved our article greatly.